# Model-Based Monitoring of Biotechnological Processes—A Review

**Velislava Lyubenova** [1,*], **Georgi Kostov** [2]  **and Rositsa Denkova-Kostova** [3]

1. Institute of Robotics, Bulgarian Academy of Science, Acad. G. Bonchev str., bl. 2, 1113 Sofia, Bulgaria
2. Department of Wine and Beer Technology, Technological Faculty, University of Food Technologies, 26 Maritza Blvd., 4000 Plovdiv, Bulgaria; george_kostov2@abv.bg
3. Department of Biochemistry and Molecular Biology, Technological Faculty, University of Food Technologies, 26 Maritza Blvd., 4000 Plovdiv, Bulgaria; rositsa_denkova@mail.bg
* Correspondence: v_lyubenova@ir.bas.bg

**Abstract:** The monitoring of the main variables and parameters of biotechnological processes is of key importance for the research and control of the processes, especially in industrial installations, where there is a limited number of measurements. For this reason, many researchers are focusing their efforts on developing appropriate algorithms (software sensors (SS)) to provide reliable information on unmeasurable variables and parameters, based on the available on-line information. In the literature, a large number of developments related to this topic that concern data-based and model-based sensors are presented. Up-to-date reviews of data-driven SS for biotechnological processes have already been presented in the scientific literature. Hybrid software sensors as a combination between the abovementioned ones are under development. This gives a reason for the article to be focused on a review of model-based software sensors for biotechnological processes. The most applied model-based methods for monitoring the kinetics and state variables of these processes are analyzed and compared. The following software sensors are considered: Kalman filters, methods based on estimators and observers of a deterministic type, probability observers, high-gain observers, sliding mode observers, adaptive observers, etc. The comparison is made in terms of their stability and number of tuning parameters. Particular attention is paid to the approach of the general dynamic model. The main characteristics of the classic variant proposed by D. Dochain are summarized. Results related to the development of this approach are analyzed. A key point is the presentation of new formalizations of kinetics and the design of new algorithms for its estimation in cases of uncertainty. The efficiency and applicability of the considered software sensors are discussed.

**Keywords:** biotechnological processes; model-based software sensors; kinetics estimation; adaptive observation; monitoring

## 1. Topicality of the Monitoring of Biotechnological Processes

Process monitoring is a mandatory element of modern industrial production. Monitoring methods also play a significant role in the study, development, optimization, and maintenance of processes in a state of maximum efficiency and desired product quality [1–3]. Contemporary biotechnological productions are no exception to the above. Biotechnological processes (BTP) are carried out in bioreactors in which microorganism growth is the result of the consumption of substrates (sources of carbon, oxygen, nitrogen, etc.). Usually, bioreactors are connected to systems for control of the physico-chemical parameters of pH, temperature, and stirring, guaranteeing suitable environmental conditions for good microbial activity. In recent decades, BTP have been at the heart of the production of a number of products primary and/or secondary metabolites. Their applications are mainly in the pharmaceutical industry, food industry, biofuel production and others. They also include classes of processes related to the treatment of wastewater from organic waste. The used scientific directions in the field of industrial biotechnology can be summarized in three

main groups: (i) microbiology and genetic engineering; (ii) biotechnological engineering; and (iii) control of biotechnological processes. Microbiology and genetic engineering aim at developing microorganisms that allow the synthesis of new products or are aimed at selecting strains to obtain the desired product or product quality. Bioengineering is aimed at improving the productivity of BTP by developing new technologies and/or improving reactor designs. Automatic control aims at increasing the productivity by developing methods for monitoring and control, leading to real-time BTP optimization. These three areas complement each other and are constantly evolving. The development of the second and the third direction is focused mainly on the study and control of BTP in real-time. In traditional biotechnological production, information on biological variables and kinetic parameters is still obtained through hardware sensors and laboratory analyses. In recent years, intensive work has been devoted to the use of available measurements for the development of so-called software sensors (SS), which provide information about the main variables and parameters of the processes in real-time. Software sensors improve BTP monitoring, which is essential for quality and quantity control of the final product, as well as for maintenance of the optimal physiological state of the culture [4]. In BTP development, the most essential information about the microbial activity is obtained by monitoring the kinetics. Kinetics knowledge has two important applications. The first one is related to the kinetic parameters included in algorithms for adaptive process productivity control [5]. The second application is related to the study of processes by continuous monitoring of the physiological state. BTP monitoring includes both methods for estimating basic kinetic parameters, such as growth rates of microorganisms, consumption of basic (limiting) substrates, production of target products, and others [6], and important non-measurable process variables.

## 2. Issues with Biotechnological Process Monitoring

The last few decades have seen the growing application of biotechnological processes in the industry [7,8], which is explained by the improvement of profitability and quality in industrial production, new legislative standards in industrial technologies, and others.

The problems arising from industrialization [9] lead to an increase in the requirements for a process monitoring (control) system, which must be equipped with a sufficient number of quality sensors to ensure stable process development in real time, optimization of the operation mode, monitoring of the physiological condition or detection of a malfunction. In practice, very few installations are currently equipped with such monitoring systems [6]. The situation can be explained by two main reasons: first, biological processes are complex because they involve living organisms whose characteristics, in their nature, are very difficult to study and understand. There are two main difficulties in modeling these processes [6,10]. On the one hand, the lack of reproducibility of the experiments and the inaccuracy of the measurements lead not only to problems related to the choice of the model structure, but also to problems related to the concepts of structural and parametric identifiability. On the other hand, difficulties arise during the validation phase of these models. The changes in the parameters are the result of metabolic changes in biomass or even genetic modifications that cannot be predicted and observed from a macroscopic (visible to the naked eye) point of view. The second main difficulty is the almost systematic lack of sensors giving access to measurements necessary for the knowledge of the internal functioning of biological processes. Most of the key variables associated with these systems (concentrations of microorganisms, substrates and products) can only be measured by laboratory analyses, which usually require heavy and expensive maintenance. The available real-time biomass and metabolite sensors are, in most cases, not robust enough for routine industrial applications. For this reason, most of the monitoring and control strategies used in industry are very often limited to indirect control of BTP by controlling the environment-specific variables, such as dissolved oxygen, temperature, pH and others. The development of so-called software sensors [11] makes it possible to overcome, to a large

extent, the discussed problems. Definition of the term "software sensor" and classification of SSs are given in the next section.

### 3. Software Sensor Concept and Software Sensor Types

By definition [9], a software sensor (SS) is an algorithm for the real-time estimation of state variables and parameters that are not measurable based on related real-time measurements that are more readily available.

Figure 1 presents the relationship between the hardware sensor and the estimation algorithm. In fact, there may be different combinations between them. Software sensors have a wide range of applications: process monitoring, control, diagnostics and prediction.

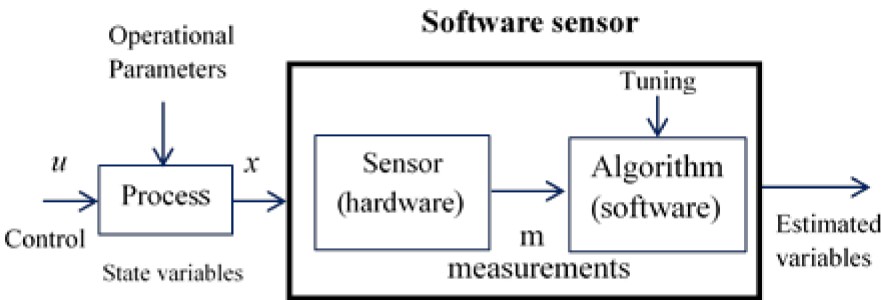

**Figure 1.** Idea of a software sensor.

The original and still predominant application field is the prediction of process variables $x$ determined at low sampling rates (laboratory analyses). Since these process variables are related to the quality of the output $y$ of the process, they are important for its control. It is essential to include additional information from higher frequency sampling, which expands the scope toward real-time prediction, in the SS structure obtained. Other important areas of application of software sensors are process monitoring [12] and malfunction detection [6]. The information obtained from the variables and parameters observed by these algorithms actually supports the process operator and allows him/her to make faster, better and more objective decisions. Software sensors are easier to maintain, as they do not get damaged mechanically and are, therefore, more efficient than hardware sensors from a financial point of view. In general, two types of software sensors can be distinguished: model-based and data-based ones. Model-based software sensors are white box estimation methods because they involve knowledge of the process. Data-based SS [13] are black box methods because they are based solely on empirical observations of the process. Combinations of model- and data-based approaches are known as hybrid ones.

Table 1 shows the basic classification of SS with some of the most commonly used methods.

**Table 1.** Basic classification of software sensors.

| Model-Based SS (White Box Type) | Data-Based SS (Black Box Type) | Hybrid SS (Grey Box Type) |
|---|---|---|
| Nonlinear observers, Kalman and Luenberger filters, Adaptive observers, etc. | Principal component analysis Least squares approach Neural networks, Neural fuzzy logics, etc. | Combinations between data-based and model-based methods: hybrid model with EKF EKF with Neural Networks, etc. |

This article discusses mainly the first-principle models as, most often, a soft sensors model-based family [14]. These models are built on a fundamental understanding of underlying physio-chemical phenomena, such as mass transfer, heat transfer and mass flow.

## 4. Model-Based Software Sensors

The most popular methods include nonlinear observers [15], extended Kalman filter [16], adaptive observers [17] and others. These software sensors most often use the abovementioned models that describe the physical and chemical basis of the process (mass balance models). The models were originally developed for planning and development of technological (production) installations and usually focus on the description of perfectly established process modes, which is one of their shortcomings in the synthesis of SS. The main properties of the most commonly used SS-based models will be briefly presented.

In the literature [17], it is accepted that software sensors for estimating state variables are defined as state observers, while those for estimating kinetic parameters of the model are defined as parameter estimators.

### 4.1. Linear Observers

Luenberger Observer (LO) includes in its structure a correction of estimates with one term, which is a function of the difference between measured and predicted outputs. The tuning parameter of this observer makes it possible to adjust the rate of convergence of the observer. The value of this parameter is a compromise between the rate of convergence and the sensitivity to disturbances that must be made.

The Kalman filter (KF) is another method of controlling this trade-off. It can be considered as a Luenberger observer with a non-stationary tuning factor. KF allows minimizing the variance of the estimation error. LO and FC require accurate information on the structure and parameters of the models. Their uncertainty can cause a large deviation in the estimates.

### 4.2. Extension of Linear Observers

The first class of observers is based on accurate information about the structure and parameters of the models. This class includes the Extended Kalman Filter (EKF), the Extended Luenberger Filter, and nonlinear observers [6].

The Extended Kalman Filter is introduced as an approximation of the optimal observer when extending the method for nonlinear systems. The process and measurement model is linearized around its estimated trajectory. This problem is equivalent to the synthesis of a Kalman Filter for a nonstationary system. Only a few theoretical results guarantee its convergence, and in the best case, the guaranteed stability is local. Estimation errors have been shown to be limited if the rank condition for the nonlinear observability is met and if the initial estimation error, as well as the modeling and measurement noise, are small enough. The application of the method leads to complex nonlinear algorithms, as well as to deviations in the estimates and even to divergence if the algorithm is not well initialized. Its stability is difficult to be proven analytically.

The values of the tuning parameters for the Advanced Luenberger Filter [18] must guarantee asymptotic stability of the linearized error dynamics. Like EKF, the stability is local and valid around the equilibrium state due to the linearized process model. It is difficult to guarantee stability in a wide range.

In the study of dynamical systems, linearization is a method for estimating the local stability of an equilibrium point of a system of nonlinear differential equations or discrete dynamical systems. Usually, a linearized approximation of the nonlinear model by a Taylor series approximation around the point of interest (equilibrium state, current estimate, etc.) is applied [17]. This is the so-called "linearized tangent model". As the linearization process leads to errors in the nonlinear system due to the calculation of the Jacobian matrix and therefore a decrease in the accuracy of the estimate, the unscented Kalman filter (UKF), central difference Kalman filter (CDKF), and square root unscented Kalman filter (SRUKF) [19] are proposed in the literature for solving this problem. These methods are based on similar ideas and belong to a class of approximate nonlinear filtering methods (sigma point Kalman filter (SPKF) method) based on Gaussian distribution. In the UKF

method [20], several sigma points for nonlinear systems to obtain second-order accuracy are used.

### 4.3. Observers for Linear Time Varying Systems

In some cases, the nonlinearity of the processes is overcome by modeling them as linear time-varying systems. In these models used in SS synthesis, linearity is achieved by considering and estimating some parameters as unknown time-varying ones [21,22]. More detail on this approach will be discussed in Sections 5 and 6.

Adaptive EKF observers can also to be applied to simultaneously estimate state variables and nonstationary parameters [23].

### 4.4. Nonlinear Observers

It is a well-known and proven fact that linear observers with constant model parameters are inadequate for estimating nonlinear processes [24]. In the case of linear systems, the problem of observer synthesis has been intensively studied and many methods are already available. In the nonlinear case, the published solutions depend on the specific problem. They are either suitable for a very limited class of nonlinear systems or their global convergence cannot be analytically proven [15].

Generally speaking, there are several approaches to observer synthesis for nonlinear systems. One is based on an extension of the linear versions of observers. Another approach known in the literature is based on the derivation of precisely linearized error dynamics [25]. As a disadvantage, these developments require appropriate transformations to be made. The existence of such a transformation imposes a series of assumptions that are difficult to verify in practice.

High Gain Observer (HGO) requires a high-quality model [26]. The idea is to present the observer in a new canonical form, i.e., as a numerical differentiator of output. This means that there exists information for the time derivatives. Then, returning to the initial coordinates, this observer will give estimates of the required variables. The synthesis of such observers requires acceptance of the hypothesis that the dynamics of the observer is faster than that of the system. The exponential stability depends on the Lipschitz condition on the nonlinear part. Variants of this method are observer with a fixed tuning factor, observer with a variable tuning factor [27], and others.

The Moving Horizon Estimator (MHE) uses nonlinear programming to solve an optimization problem that increases computational costs [28]. The method includes some restrictions and calculates state trajectories based on the process model and the initial state vector. With a large deviation from the initial estimates from the true values, two problems, related to falling into the local optimum and a lack of convergence, appear.

Observers, based on Lipschitz systems, ensure convergence estimates by selecting the value of the observer tuning parameter so that the Lipschitz constant meets a specific condition. The Lipschitz constant is the maximum ratio between variations in the output space and variations in the input space of a function and thus, is a measure of the sensitivity of the function with respect to input perturbations [29]. A number of global optimization algorithms rely on the value of the Lipschitz constant of the objective function [30]. Various approaches to its estimating are known, including those based on a priori knowledge of the particular process, the Lipschitz optimization without the Lipschitz constant, etc. [31–33]. The solutions, obtained independently of the prediction task, show a trend for noise sensitivity.

Linear matrix inequality (LMI) is another approach to observer synthesis [34]. The main idea is to prove the inertia of the nonlinear part of the error dynamics and to satisfy the Kalman–Popov lemma for the linear part by solving a task related to a linear matrix inequality.

According to the type of uncertainty of the process, observers are divided into two classes. The first class deals with cases in which uncertainties are due to noises in measurements or modeling. When the BTP model is presented based on mass balance equations, it

includes a term representing the kinetics, which in many cases cannot be modeled. For this reason, observers with unknown inputs [35] are used. Their principle is the exclusion of the unknown part of the model by the appropriate transformation and introduction of an auxiliary variable. These observers define the second class (Asymptotic Observers) and are often used in BTP [6,10], replacing the knowledge of the missing part of the process model with available measurements. The main advantage of this method is that it does not depend on the process kinetics. The main disadvantage is that the rate of convergence depends only on the experimental conditions of the process. A necessary condition is that the number of measurements is equal to the number of unknown kinetic reactions. Another disadvantage is that convergence in batch processes is not guaranteed.

The nonlinear estimation methods discussed above are compared in Table 2 in terms of tuning parameters and types of stability. An important problem in the synthesis of observers and estimators is to prove their stability by analyzing the dynamics of errors. The stability can be local, global, exponential or asymptotic depending on the method and the structure of the model [36–38].

**Table 2.** Critical analysis of nonlinear methods.

| No | Observer | Number of Tuning Parameters | Stability |
|----|----------|-----------------------------|-----------|
| 1 | Extended Kalman Filter | Two (R,Q) | Local |
| 2 | Extended Luenberger Observer | Number of the poles (ordered by the system) | Local |
| 3 | Linearization of the error | Depends on the linear method | Global |
| 4 | Accurate linearization | Depends on the linear method | Local or global |
| 5 | High gain observer | Number of the poles | Local or global |
| 6 | Moving horizon observer | One | Asymptotic |
| 7 | Linear matrix inequality | Two | Global |
| 8 | Based on inertia | Number of states | Global |
| 9 | Asymptotic Observer | Depends on experimental conditions of the process | Asymptotic |

The next subsection will review the model-based SS for biotechnological processes, presented in the literature, as they are of particular interest in this article. An up-to-date overview of the data-driven SS is presented in [20,39,40].

## 5. Model-Based BTP Software Sensors

In the last decades, software sensors have been intensively developed and widely used for BTP monitoring. A number of review publications related to the topic have been proposed in the literature [6,14,24,28,41–43].

As the estimation of process kinetics requires real-time measurements of state variables, many of the proposed methods are preceded by the synthesis of appropriate observers or adaptive algorithms proposed for the simultaneous evaluation of parameters and variables. Two classes of state variable observers are most often applied [6,44]: classical nonlinear observers based on accurate knowledge of the model structure, and asymptotic observers in which the synthesis is based on mass and energy balance, and knowledge of process kinetics is not required. Regarding the estimation of process kinetics, a widely used method is based on the theory of linear systems [17]. This method includes estimating the immeasurable state variables with asymptotic observers; then, the measured and/or estimated variables are included in the synthesis of kinetic estimators, the so-called observer-based estimators [17]. The most commonly used model-based methods for BTP estimating are the following.

Methods based on balance equations: These methods are based on theoretical or experimentally derived relationships between the measured variables and the parameters to be evaluated. For the BTP purposes, various nonlinear empirical/balance equations that do not take into account measurement noise and model uncertainties are derived.

Methods based on the advanced Kalman filter: The extended Kalman filter (EKF) is a standard nonlinear method used to simultaneously estimate the state variables and

parameters of nonlinear systems. The estimation is performed by linearizing the nonlinear model around the current estimation and then applying the Kalman filter. Thus, it can be considered an approximation of the optimal observer when extending the method to nonlinear systems. The linearization of dynamic process models can be accomplished with the Taylor series around some point of equilibrium. This idea was developed in [17]. A number of developments related to the application of EKF for biotechnological processes have been published in [12,22,34,42,45–51]. The results were satisfactory, but the uncertainty in the model parameters can generate a large deviation in the estimate [2]. The application of EKF to estimate specific growth rate was performed in [17,47,52,53]. Despite the fact that this method is easy to apply, it has two main drawbacks. First, its stability and convergence are local and there is no guarantee that acceptable estimates will be obtained over a wide range of operating conditions. Second, the exponential convergence of the filter is based on the assumption that the linearized model is observed close to the equilibrium state. As noted in [54], this assumption is not valid in many practical cases.

Probabilistic observers [55] are considered for a class of continuous biological processes. In comparison to classical open loop asymptotic and interval observers, the method provides information on the confidence level of the estimates rather than simple upper and lower bounds. Moreover, unlike Kalman filters, probabilistic observers are not restricted to Gaussian distributions for the uncertain parameters.

Methods based on estimators and observers of deterministic type: By definition, observers estimate state variables in deterministic linear systems based on knowledge of the mathematical model of the process and available measurements. Most real systems often involve uncertainties resulting from non-stationary parameters and noises. In addition, linear observers with constant parameters do not lead to good characteristics for nonlinear processes, such as BTP. Therefore, nonlinear and adaptive observers that are robust to uncertainties and are adaptable to the process nonstationarity are more suitable for estimating the parameters and variables of biotechnological processes.

Various approaches have been applied in the literature to estimate the kinetics of the biomass growth, substrate consumption and production of BTP products. Most of them apply or further develop the method proposed by Basten and Dochain, based on the theory of linear time-varying systems. The method involves the synthesis of observer-based estimators (OBE), which are part of the general theoretical methodology developed by the two authors in [17]. This methodology is based on the General Dynamic Model (GDM) of the process derived from the general reaction scheme, a qualitative description of the main metabolic reactions. The essence of this methodology, as well as the main types of state observers and kinetic estimators, developed on the basis of GDM, are given in the next subsection, as they are one of the bases for its further development presented in the article.

The GDM approach has proven its successful applications for a wide variety of BTPs in the food, pharmaceutical and environmental industries, more precisely in the monitoring and control of these processes, some of which are presented in [48,56–61]. The stability of the overall structure of OBE is successfully analyzed in [1]. The tuning of such estimators requires special attention, as the parameters are non-stationary. This problem is investigated in [60,62,63], where a tuning strategy independent of the values of the state variables is proposed. In [62], an approach for tuning the general structure of OBE is presented.

Compared to other approaches (EKF, nonlinear methods), observer-based estimators have the following advantages: they are characterized by a simple structure, there is no need to introduce nonlinear modeling (or black box type) of unknown parameters, and the stability analysis does not lead to rigid tuning rules (as with the H∞ method). In cases where many reaction rates are included, the application of the approach requires the tuning of many parameters. This problem is partly solved by using the method of the nonlinear systems theory, High gain observer (HGO) [27,64–71], and by analyzing the relationship between the stability of the OBE and its tuning parameters. In the HGO method, the tuning expression includes only one parameter, regardless of the number of components (state variables) and reactions. The first developments of the method were

published in the late 1980s; later, further development of the method was proposed by French researchers (Gauther, Hammori, Farza) [72]. The proposed estimation schemes have another advantage: they are robust in terms of model perturbations, as they do not require a kinetic model. In [71], a simple high gain continuous discrete time observer is proposed to handle the estimation problem of reaction rates in bioreactors, using delayed sampled measurements of the component concentrations in the context of multi-rate sampling of the outputs, each one of which is affected by the constant delay. The adaptive version of the high gain observer for the strictly triangular systems subjected to constant unknown disturbances is proposed in [64] and has been applied to a continuous culture of *Spirulina maxima*. One of the limitations of using the high tuning factor method is that it requires measurements of state variables to estimate the respective reaction rates, which, in many cases, cannot be realized in real conditions. In addition, the yield coefficients are considered constants, which is acceptable only for continuous processes and for some batch-feeding processes. Another problem present in these observers is that changes in their dynamics (derivatives) are considered disturbances, i.e., the estimation error converges to a limited area around zero. In exponential observers (EO) (the rate of convergence can be freely selected in the synthesis by the tuning parameters), including the HGO, fast convergence leads to high sensitivity to measurement noise or other disturbances.

A comparative review of multi-rate moving horizon estimation schemes for bioprocess applications is given in [28]. It is emphasized that moving horizon schemes outperform the constrained, extended Kalman filter in dealing with the challenges that are usually encountered in bioprocess operations. Smoothing arrival cost update approaches are superior in recovering from inaccurate initial conditions and covariances.

In sliding mode observers [73–77], the idea is to bring the system into sliding mode by discontinuous action on a subspace for which the estimation error is zero. This leads to their unique properties, as the ability to generate a sliding motion on the error between the measured output and the observer output ensures that the observer in sliding mode produces a set of state estimates that are exactly commensurate with the actual system output [73]. These SS have better characteristics in terms of the rate of convergence and robustness of disturbances compared to the exponential ones. Convergence is achieved within a finite time, which is important because the processes are of limited duration [74]. Estimators in first and second order sliding mode have been discussed in the literature. The first-order ones were developed in [77] for the purpose of estimating the specific growth rate from real-time measurements of biomass concentration. They were based on the high tuning method with the addition of a corrective, discontinuous member. The obtained estimates were robust under typical model uncertainties, as a global convergence was proven on the basis of the Lyapunov stability theory [36] and the concept of systems with variable structure. In [73], a SS in a second-order sliding mode of the specific growth rate was proposed. It was a modified version of a "moving in a spiral" algorithm. In addition to the advantages of first-order sliding estimators, this SS gave smooth estimates. The results of [73] are summarized in [76]. An algorithm in second-order sliding mode has been proposed, estimating $p$ specific kinetic rates of production or consumption, using $p$ related real-time measurements of state variables. Second-order sliding mode SS give more accurate estimates than first-order ones, and their advantage is in the fact that there is no error under limited changes in the evaluated variables.

The kinetics estimation methods proposed above assume that all state variables necessary to estimate the respective reaction rates have been measured. In many practical situations, only some of the state variables are available. Therefore, it is necessary to develop SS for the immeasurable state variables. The asymptotic observers (AO) proposed in [17] are the most commonly used method to solve this problem, as it does not require knowledge of the kinetics of the process. As mentioned, a disadvantage of AO is that the rate of convergence is limited by the experimental conditions and that these observers are not applicable in batch cultivation.

Hybrid software sensors as a combination of model- and data-based methods are a promising direction in this area. It is in the initial stage [20] and therefore, not many articles have been reviewed. Hybrid models emerge as a timely pragmatic solution for synergistically combining available process data and mechanistic understanding. A new application of the hybrid-EKF method is presented in [78], i.e., hybrid models combined with an extended Kalman filter for real-time monitoring in mammalian cell culture processes. It was demonstrated that for industrial use, the application of a hybrid EKF as a soft sensor shows a 50% improvement in prediction accuracy, compared to the most modern soft sensor tools. In the literature, there are hybrid SS, which combine the advantages of exponential and asymptotic observers [79,80]. A parameter related to the confidence in the quality of the kinetic model governs the combination of these two classes of observers, i.e., an exponential observer with a high-quality model and an asymptotic observer for an unknown kinetic model. Different approaches for defining and estimating this tuning parameter have been proposed, and different exponential observers have been considered. In [80], a hybrid asymptotic extended Luenberger observer was proposed, which was validated by simulations of a batch feeding bacterial culture. A novel hierarchical EKF/MHE approach was presented in [81] for process monitoring in airlift bioreactors. The complementary properties of the two widely used methods combined the fast estimation of EKF and MHE's optimal performance. In [82], an interval sliding mode observer design method for uncertain systems is proposed. Uncertainty was assumed between known minimum and maximum values. The observer was then constructed via convex weighted sum of an upper estimator corresponding to the maximum value of the uncertainty and a lower estimator corresponding to the minimum value of the uncertainty. The weighting factor was calculated at each time point from the different measured outputs and the bounds of the interval of the estimator. In [79], a new stable nonlinear observer for BTP state variables was proposed, which combined a hybrid asymptotic extended Luenberger observer (using a new definition of the confidence parameter of the kinetic model), tuned on the basis of a stable H∞ approach and differential-algebraic representation.

The observability analysis of bioprocess models is a valuable tool for the development of mechanistic soft sensors [83]. It can provide an indication of the possibility and reliability of SS estimations by analyzing the structural properties of first-principle models. In the paper, the applicability of the observability analysis is demonstrated for two classes of upstream bioprocesses.

To solve the problem with the lack of measurements of state variables in the literature, combinations of estimators of kinetic parameters and observers of state variables have also been proposed. They are the so-called adaptive observers [6,17,24,53,65,84–91]. A well-known approach is to consider the parameters as additional state variables without a model for their dynamics and evaluate them with an extended Kalman filter, a Luenberger observer or other type of observers. The extension of the condition was limited to such a number of parameters at which the observability conditions were satisfied by the available measurements. In these studies, all (or most) of the estimated parameters were considered constants. In [84], a systematic approach is proposed for simultaneous estimation of the state and non-stationary parameters of nonlinear systems. An adaptive observer was synthesized by optimizing a linear matrix inequality. The approach was demonstrated on the basis of a real model of the wastewater treatment process.

Some authors propose kinetics estimation approaches based on measurements of available rates (rate of oxygen consumption, $CO_2$ release, concentrations of dissolved oxygen in the culture medium, $CO_2$, etc.) [92]. In [93], the SS of specific growth rates, substrate consumption and product production are proposed, based on real-time measurements of the oxygen consumption rate. They were an extension of the approach in [17], in the specific case when only this rate was measured. In [94], a new structure of an adaptive observer of biomass concentration and its growth rate in the presence of measurements of oxygen consumption rate is proposed.

The development of cascading structures by the SS is also a promising area in BTP monitoring [58,95–97]. They are applied in cases when the processes are characterized by several growth rates (multi-rate), with several microorganisms or when the estimates from one SS are used as input for the subsequent ones. The proposed cascades of software sensors for the rates of production and consumption of an intermediate metabolite with input information on the concentrations of an external carbon source and the intermediate metabolite in the culture medium deserve special attention. The new element in the synthesis is that the difference between these estimated rates is accepted as a key parameter for monitoring the physiological states of processes described by one model and characterized by the growth of one or two microorganisms. The structures are applied for the processes of the production of biopolymers from mixed cultures of *L. delbrueckii* and *R. eutropha* [96], and the simultaneous saccharification and fermentation of starch to ethanol from a recombinant *S. cerevisiae* strain YPB–G [58].

A similar approach is proposed for monitoring the current physiological state of a class of biotechnological processes characterized by the production and consumption of an intermediate metabolite, as well as two switching sub-models describing three physiological states. It is based on the introduction of an adaptive key parameter (marker) to recognize the current physiological state [98,99].

Table 3 provides a critical analysis of the most commonly used software sensors for BTP, highlighting their main advantages and disadvantages.

**Table 3.** Comparison of commonly used methods in BTP monitoring.

| Method | A Priori Information | Advantages | Disadvantages |
|---|---|---|---|
| Balance equations | Input–output connections | Simple calculations based on approximate models | There are no reliable estimates in the presence of uncertainty |
| Extended Kalman Filter | Mathematical Model (MM) | Good results in stochastic disturbances and measurement noise | Accurate process models; problems at inaccurate initial estimates and covariance matrices. |
| Hybrid observers (EO+AO) | MM | Exact estimates for deterministic nonlinear processes | Exact model of the exponential observer; limited AO convergence rate. |
| Observer-based estimator | MM; On-line measurements related to the estimated rate | Simple linear structure; robustness; possibility for optimal tuning | A large number of tuning parameters; estimates depend on changes in rates, constant yield coefficients |
| High gain SS | MM; On-line measurements related to the estimated rate | One tuning parameter; effective work with nonlinear processes; robustness | The estimates depend on changes in rates; constant yield coefficients; the exponential stability depends on the Lipschitz condition. |
| Sliding mode SS | MM; On-line measurements related to the estimated rate | Smooth estimates for second-order systems, without errors at limited changes in estimated variables | First-order methods-effect of rapid change of estimates until entering the sliding plane; constant yield coefficients. |

The following subsection presents the basic concepts and essence of the General Dynamic Model (GDM) method for the purposes of SS synthesis. This is, on the one hand, a widely used and modified method used in the literature. On the other hand, the GDM approach is the basis for the further development of the theory, presented below.

## 6. General Dynamical Model Approach and its Further Development for SS Synthesis

The approach is based on deriving a general dynamic model of BTP in an ideal-mixing reactor based on reaction schemes [17]. The model is presented in a vector-matix form with the following differential equation:

$$\frac{\mathrm{d}\xi}{\mathrm{d}t} = \mathbf{K}\boldsymbol{\varphi}(\xi, t) - D\xi + F - Q \tag{1}$$

$\xi$—vector of concentrations of components dissolved in the nutrient medium;
$\mathbf{K}$—a matrix of yield coefficients;
$\boldsymbol{\varphi}$—a reaction rate vector;
$D$—dilution rate;
$F$—a vector of feed rates;
$Q$—flow rates of gaseous components from the reactor.

*Idea for minimal modeling of reaction rates*

It consists in representing the vector of reaction rates $\boldsymbol{\varphi}(\xi, t)$ as a product of a matrix $\mathbf{H}(\xi)$ of known functions of the state variables $\xi$ (t) and a vector $\boldsymbol{\rho}$ (t) including completely unknown nonstationary parameters as follows:

$$\boldsymbol{\varphi}(\xi, t) = \mathbf{H}(\xi)\boldsymbol{\rho}(t) \tag{2}$$

This allows different types of process uncertainties to be taken into account and a wide range of practical situations to be covered.

Another problem with BTP monitoring and control is the lack of reliable sensors for real-time measurements of state variables. For this reason, software sensors for these variables are being developed.

Tables 4 and 5 give a brief description of the different types of state observers and parameter estimators developed in [17], depending on the available a priori process information. The presented SS are diverse both in terms of the methods used and the a priori information used.

GDM is described by a system of ordinary differential equations in which parameters and variables depend on the sole time variable since it has been developed for stirred tanks bioreactors. Its extension in terms of the spatial change of the state variables has not been thoroughly studied in the general case yet. In [100,101], an extension related to biochemical tubular reactors is presented. An observer of state variables based on GDM, using partial–differential equations is derived. It is applied to gluconic acid production process.

**Table 4.** State observers.

| $\frac{d\xi}{dt}=\mathbf{K}\boldsymbol{\varphi}(\xi,\mathbf{t})-D\xi+F-Q$ | A Priori Information | |
|---|---|---|
| **Software sensors** | $\boldsymbol{\varphi}(\xi, \mathrm{t})$ | Matrix **K** |
| **Exponential and asymptotic observers** <br> *Limitation*: Nonlinear models | Known | Known |
| **Asymptotic observers with auxiliary variables** <br> *Advantages:* Simple structure in comparison with <br> EKF and ELO and independence from $\boldsymbol{\varphi}(\xi, \mathrm{t})$; <br> *Limitations*: Limited rate of convergence; conditions <br> for matrix inversion. | Unknown | Known |
| **Adaptive observers** <br> Extended Observers of Kalman and Luenberger <br> *Limitations*: Nonlinear structure <br> Asymptotic observers <br> *Constraints:* Structural identifiability | Unknown | Unknown |

**Table 5.** Parameter estimators.

| Kinetics $\varphi(\xi,t) = KH(\xi,t)\rho(\xi,t)$ | Estimators of $\rho(\xi,t)$ and K |
|---|---|
| Observer-based estimator of $\rho(\xi,t)$ *Limitations:* The disturbance vector includes quite a few members resulting from considering the kinetics a product of three members | Estimation of $\rho(\xi,t)$ with known K |
| Conditions: (1) Need for **z** transformation, in which the dynamics is independent of economic coefficients; (2) Reformulation of kinetics $\varphi(\xi,t) = \mathbf{KH}(\xi,t)\rho(\xi,t) = \mathbf{\Phi}(\xi,\mathbf{Z})$, so that $\mathbf{\Phi}(\xi,\mathbf{Z})$ be independent of yield coefficients | Estimation of $\rho(\xi,t)$ independently from K |
| Case 1: Complete measurements of state variables Reformulation of kinetics $\varphi(\xi,t) = \mathbf{KH}(\xi,t)\rho(\xi,t) = \mathbf{\Phi}(\xi)\theta$ The estimation of $\theta = f(\alpha,\kappa)$ and **f** to be invertible, i.e., $\begin{vmatrix} \boldsymbol{\alpha} \\ \mathbf{k} \end{vmatrix} = \mathbf{f}^{-1}(\theta)$ Case 2: Incomplete measurements of state variables *Constraints*: Condition for invertibility of matrix and constantly stimulating $\mathbf{\Phi}(\xi)$ at unknown yield coefficients | Simultaneous estimation of $\rho(\xi,t)$ and **K** |
| *Limitations:* Structural identifiability of yield coefficients from function f i.e., $\theta = f(k)$ | Estimation of **K** independently from $\varphi(\xi,t)$ |

Usually, the software sensors are designed using operational models with constant yield coefficients. For many industrial biotechnological processes, such as wastewater treatment and processes in inhomogeneous mediums, reproducibility is poor [50]. Hence, the assumption of a constant yield coefficient leads to inexact results because of considerable changes in these parameters during a process or within different production batches. This change is due to adaptations of the metabolic pathways, protein expression pattern, and random mutations of organisms, as well as the occurrence and dynamics of population heterogeneities in single species, especially in multispecies bioprocesses [102,103]. For cases in which the process kinetics is completely unknown, or the yield coefficients are not constant, two generalized approaches for SS synthesis are proposed [96,104]. They can be considered supplementary to the theory of Bastin and Dochain and are based on two new formulations of BTP kinetics presented in Table 6.

**Table 6.** New formalizations of kinetics $\phi(t)$.

| | Process Kinetics Formalization |
|---|---|
| $\phi(t)$ fully unknown time-varying parameter | $\boldsymbol{\phi_m}(t) = Y(t)\varphi(t)$ $\boldsymbol{\phi_m}(t)$—vector of known kinetics; $\varphi(t)$—key kinetic parameter, which describes the dynamics of the main state variables; $Y(t)$—vector of yield coefficients comprising remaining parts of the state variables' kinetics. |

The two general structures of SS are briefly described in Table 7.

In the first generalized SS, the input is the state variable whose kinetics $\phi(t)$ is estimated. The originality in the first SS is (i) considering kinetics as a vector of completely unknown time-varying parameters, thus avoiding estimation errors arising from constant values of yield coefficients or other kinetic parameters; and (ii) the linear structure of this SS makes it possible to propose an optimal tuning of the parameters based on an analytically derived asymptotic upper limit of the error in the estimation.

**Table 7.** General SS based on the new formalizations from Table 7.

| General SS | |
|---|---|
| **General SS of $\phi$(t)**<br>The asymptotic upper limit of the estimation error is derived, which is the basis of the proposed optimal tuning. The advantage of the new software sensor is that it provides reliable kinetic information when the kinetics models are unknown or inexact ones.<br>**Disadvantage:**<br>The effect of measuring noise cannot be completely ruled out.<br>**Applications:** [58,104–106]. | **General SS of $Y$(t) and $\varphi$(t)**<br>A linear structure of a generalized software sensor of 4th/5th order is derived. The input is the measurable kinetics and a simultaneous estimation of both parameters at the output is achieved. An analysis of the stability of the obtained structures is performed and original tuning procedures for processes taking place in an inhomogeneous/homogeneous environment are proposed.<br>**Advantage:** The tuning is reduced to selecting two parameters for processes that are realized in an inhomogeneous environment, while for the same processes in a homogeneous environment only one parameter is needed.<br>**Disadvantage:** Only local asymptotic stability can be proven.<br>**Applications:** [107–109]. |

The new formalization of the kinetics was used in the synthesis of discrete versions of SS based on the stability analysis of the following processes: continuous fermentation with immobilized *Saccharomyces cerevisiae* strain [106] and production of $\alpha$-amylase by *Bacillus subtilis* [105]. The method has been used in the synthesis of SS in the following processes: production of gluconic acid by *A. niger* strain [104]; and the synthesis of biopolymers by mixed cultures of *L. delbrueckii* and *R. eutropha* and the simultaneous saccharification and fermentation of starch to ethanol by a recombinant *S. cerevisiae* strain YPB–G [58].

The second structure is characterized by a ratio $\phi m$(t) =$Y$(t). $\varphi$(t) between measured and estimated parameters, which provides the synthesis of asymptotically stable SS. The originality of this approach consists in its (i) presentation of the process kinetics with two unknown time-varying parameters with physical meaning, (ii) derivation of a linear structure of the SS using logarithmic transformations of the parameters that facilitate stability analysis, and (iii) derivation of stable fourth- and fifth-order structures of the SS, satisfying conditions for asymptotical stability. The proposed tuning procedures lead to a reduction in the number of SS parameters in a different number of measurements available in industrial practice. The method was applied for monitoring the kinetics of classes of processes realized in an inhomogeneous [103,109]/homogeneous [107] environment. The results were used to study the kinetics of batch fermentation with strains of *B. subtilis* and *E. coli*, as well as to monitor the denitrification phase in the process of wastewater treatment with activated sludge [108].

## 7. Discussion

The model-based SS discussed above have their advantages and disadvantages, which were considered in the previous sections. It should be noted that the choice of SS method for a specific process and in particular BTP is not an easy task and requires an analysis of the following a priori information:

- Complexity of the specific process;
- Full/partial knowledge of the model structure;
- Available process information (quality and quantity of available offline and online measurements), the types of noises and uncertainties, etc.

On this basis, it is necessary to assess which model and model-based SS will be most adequate. It should be borne in mind that the model should describe the dynamics of the process as accurately as possible and on the other hand should not be too complex to allow the application of the specific SS.

Below are some guidelines for selecting SS-based data that do not claim to be exhaustive.

It is not difficult to conclude that in a nonlinear system containing Gaussian noise, UKF and CKF would give better results than EKF, as they do not require a large number of calculations. EKF is a good solution if the nonlinearity of the system is not strong, and the process model is accurate [20].

Regarding the uncertainties in the model (especially in the modeling of process kinetics), a common problem for BTP, a number of methods have been proposed in the literature that successfully solve it. The OBE method as part of the GDM approach is very often successfully applied for monitoring BTP processes, as the unknown part of the kinetics is estimated as an unknown time-varying parameter. Although not a statistical method, it has the ability to filter measurement noise to some extent [17]. The asymptotic observers that exclude unknown kinetics when estimating unmeasured variables are a good solution when the dilution rate, determining convergence rate, does not have very low values.

The observer-based estimators mentioned above are characterized by simplicity of design, good convergence and stable properties. In some cases, when many reactions are involved, it is necessary to calibrate many parameters. To overcome this problem, high-gain observers that include one tuning parameter may be recommended [44]. The HGO in general has the advantages of OBE, but the amplification of noise through its own phenomenon of amplification and peak of the observer are its disadvantages [110]. Usually, when tuning the parameters of much of the deterministic SS-based model, a trade-off between the convergence rate and the sensitivity to noise measurement has to be made.

Sliding mode SSs have robustness to disturbances, insensitivity to unknown inputs and a finite time convergence. Their disadvantage is the chattering effect, increasing the systems relative degree and, in some cases, destabilizing the closed loop system [110]. The chattering phenomenon is still present even in the currently developed higher-order sliding mode (HOSM) observers.

In cases where the whole dynamics is unknown and the noise from the measurements is not great, the software sensors proposed in [104], which consider and estimate the kinetics as completely unknown parameters, are recommended. They have the properties of OBE, but avoid estimation errors resulting from constant values of yield coefficients, other kinetic parameters and some measured variables.

For cases where the kinetics of the process are completely unknown and the yield coefficients are time-varying, it is recommended to apply the approach proposed in [96]. Its advantages and disadvantages are described detail in Section 7.

For software sensors design, different software packages (numerical schemes) are used. They are different depending on the used method and software environment. However, there are some modules that are common to all cases:

- A process database creation module.
- Module containing programs that solve the differential equations of SS and/or model used.
- Module containing programs for tuning of SS parameters and/or model identification.

One promising direction in the field, commented in recent years in a number of articles, is the development of hybrid SS [20]. The idea is to combine the strengths of both data-based [20,39,40] and model-based methods for software sensors design. The authors recommend publications [14,20], where this issue is discussed in more detail.

## 8. Conclusions

Model-based software sensors are a common contemporary monitoring method in the biotechnology industry. The models used in SS synthesis must be (i) as accurate as possible to mimic the basic characteristics and dynamics of the process, and (ii) simple enough to monitor and control. The approaches based on the General Dynamic Model approach are widely applied simultaneously with those for nonlinear systems, such as extended Kalman and Luenberger filters, moving horizon, high-gain approach, sliding mode observers, interval SS, cascade structures, joint estimation of state variables and parameters, etc. The proposed SS structures depend on the available input information and

the expert's requirements for the observed parameters and variables for each case under consideration. The analysis and comparison of the most commonly used methods show that nonlinear and/or adaptive methods, which are robust to uncertainty and non-repeatability of experiments under the same conditions and are adaptable to the non-stationarity of the process, are more promising and reliable in terms of the inherent complex nature of biotechnological processes. We believe that the development of hybrid SS, as a future direction in the field, will be able to synergistically combine the advantages of model-based and data-based methods.

**Author Contributions:** Conceptualization and methodology, V.L., G.K.; performed the literature search, V.L.; G.K., R.D.-K.; original draft preparation, V.L., G.K.; review and editing, V.L., G.K., R.D.-K. All authors provided critical feedback and helped shape the research, analysis and manuscript. All authors have read and agreed to the published version of the manuscript.

**Funding:** The research in the present manuscript was funded by the National Scientific Fund of Bulgaria, Grant KII-06-H32/3 "Interactive System for Education in Modelling and Control of Bioprocesses (InSEMCoBio)".

**Institutional Review Board Statement:** Not applicable.

**Informed Consent Statement:** Not applicable.

**Data Availability Statement:** Not applicable.

**Conflicts of Interest:** The authors declare no conflict of interest.

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
