# Peer review of "Model-Based Monitoring of Biotechnological Processes—A Review"

_processes, doi:10.3390/pr9060908_

Round 1

Reviewer 1 Report

Dear authors,

Presented review paper has very strong potential to be a very interesting for worldwide readers community of MDPI Process journal, but in present form it has a lot of issues that should be solved before it became acceptable. Those issues are:

  1. Too much self-citation, I found 19 citations of the first author papers, I can suppose that first author’s work is significant in the field of the software sensors, but goal of review paper to represent level of research all-over the world;
  2. Please add additional keywords, total at least five keywords usually common for the review papers;
  3. The abstract is too short and does not represent main statements of the paper;
  4. Line 34. “The used scientific directions in the field of biotechnology…” Please change to the: “The used scientific directions in the field of industrial biotechnology…” Because term “biotechnology” includes much more than discussed in this sentence;
  5. Paragraph II line 59, please change in the paragraph titles word “problems” to “issues”
  6. Paragraph II, the statements in this part of paper is absolutely correct, but it is necessary to expand this paragraph with some references on the papers with scientific data concerning those issues.
  7. Lines 90-92. Usually definition requires reference until this definition developed by authors.
  8. Please discuss necessity of the figure 2.
  9. Information on the Figure 3 can be presented in the table, and I suppose table will be more suitable for readers;
  10. Lines 432, 443 and etc. All formulas should have its own numbers like (1);
  11. The statistical approaches and especially artificial neural networks based software sensors does not discuss properly in the paper, but there are a lot of researches and applications in this field in the recent years;
  12. It would be significant add in the paper if authors at least breathily discussed issues related with software sensors in case of application cells with synthetic genetic circuits as sensors for biotechnology processes.
  13. The discussion section is necessary, author should discuss concerning efficiency of presented sensors or perspectives of some new approaches’ with respect to existing one.
  14. Author should check spelling all over the paper.

Author Response

Dear Editor,

We would like to thank all reviewers for their very valuable comments and suggestions. Taking into account all of them, we believe that the quality of the article has been improved.

The “Track Changes” function is used during the paper revision process. Additionally, all improvements are presented in red colour in order to be easily visible to the editors and reviewers in “Simple markup” mode.

Following are the detailed answers, point by point of each of the reviewers.

Answers to Reviewer 1

  1. Too much self-citation, I found 19 citations of the first author papers, I can suppose that first author’s work is significant in the field of the software sensors, but goal of review paper to represent level of research all-over the world;

We agree with the remark made and the number of publications of the first author is significantly reduced in the cited literature.

Other literature sources are included.

The larger number of publications of the first author included in the literature is to present the latest investigations related to the development of the General Dynamical Model Approach. In this regard, specialists from Institute of Robotics in the field of bioengineering are a team that has significantly contributed to the development of GDM, reflected in a number of publications in prestigious specialized editions.

  1. Please add additional keywords, total at least five keywords usually common for the review papers;

The number of keywords has been increased as follows:

Key words: biotechnological processes, model-based software sensors, kinetics estimation, adaptive observation, monitoring

  1. The abstract is too short and does not represent main statements of the paper;

The abstract was modified and expanded as follows:

The monitoring of the main variables and parameters of the biotechnological processes is of key importance for the research and control of the processes, especially in industrial installations, where there is a limited number of measurements. For this reason, many researchers are focusing their efforts on developing appropriate algorithms (software sensors (SS) to provide reliable information on unmeasurable variables and parameters, based on available on-line information. In the literature, a large number of developments related to this topic that concern data-based and model-based sensors are presented. Up-to-date reviews of data-driven SS for biotechnological processes have already been presented in the scientific literature. Hybrid software sensors as a combination between the above mentioned ones are under development.This gives a reason the article to be focused on a review of model-based software sensors for biotechnological processes. The most applied model-based methods for monitoring the kinetics and state variables of these processes are analyzed and compared. The following software sensors are considered: Kalman filters, methods based on estimators and observers of deterministic type, probability observers, high-gain observers, sliding mode observers, adaptive observers, etc. The comparison is made in terms of their stability and number of tuning parameters. Particular attention is paid to the approach of the General Dynamic Model. The main characteristics of the classic variant proposed by D. Dochain are summarized. Results related to the development of this approach are analyzed. A key point is the presentation of new formalizations of kinetics and the design of new algorithms for its estimation in cases of uncertainty. The efficiency and applicability of the considered software sensors are discussed.

  1. Line 34. “The used scientific directions in the field of biotechnology…” Please change to the: “The used scientific directions in the field of industrial biotechnology…” Because term “biotechnology” includes much more than discussed in this sentence;

This change is reflected in the text

  1. Paragraph II line 59, please change in the paragraph titles word “problems” to “issues”

This change is reflected in the text

  1. Paragraph II, the statements in this part of paper is absolutely correct, but it is necessary to expand this paragraph with some references on the papers with scientific data concerning those issues.

This paragraph was extended with some references on the papers with scientific data concerning those issues.

  1. Lines 90-92. Usually definition requires reference until this definition developed by authors.

A reference was added to the definition

  1. Please discuss necessity of the figure 2.

Figurе 2 is excluded from the article. The following sentence has been added: Software sensors are used for process' monitoring, control, diagnostics and prediction.

  1. Information on the Figure 3 can be presented in the table, and I suppose table will be more suitable for readers;

The figure has been replaced by a table.

  1. Lines 432, 443 and etc. All formulas should have its own numbers like (1);

The numbering of the formulas is introduced in the text.

  1. The statistical approaches and especially artificial neural networks based software sensors does not discuss properly in the paper, but there are a lot of researches and applications in this field in the recent years;

The article discusses in detail statistical methods based on models, as Extended Kalman Filter, Probabilistic observers, Moving Horizon Estimator (MHE), etc. as presented in the literature. Consideration of the neural network approach to SS synthesis is beyond the scope of this article, as it is a date-based method according to the widely used classification.

Some up-to-date reviews of data based SS has already been given [95,103,107].

  1. It would be significant add in the paper if authors at least breathily discussed issues related with software sensors in case of application cells with synthetic genetic circuits as sensors for biotechnology processes.

Such application is very interesting in opinion of the authors. Moreover, we intent to  develop a software sensor design including appropriate models and concomitant parameters for the estimation of immeasurable state variables and metabolic fluxes at non-steady state

But such software sensors are based on structural models and are not the subject of this article.

  1. The discussion section is necessary, author should discuss concerning efficiency of presented sensors or perspectives of some new approaches’ with respect to existing one.

Such discussion section is included.

  1. Author should check spelling all over the paper

It is realized.

Reviewer 2 Report

The article is an interesting review of advanced mathematical models for the dynamical control of biotechnological processes (BTP). In the first part of the article, the authors provide a wide overview of BTP illustrating them through the support of numerous examples. In the second part, the authors describe the principal existing approaches to design software-based sensors of BTP. In the third part, the authors analyze a general approach, denoted as GDM, based on a system of reaction equations to mathematically model the kinetics of an ideal-mixing reactor. The article is of interest and certainly deserves to be considered for publication. Before this, the authors should successfully address the questions listed below:

  1. To this Reviewer's opinion, the most significant aspect of this manuscript is the acknowledgement that mechanistic-based (MB) approaches may be effectively adopted to design software-sensors (SS) for BTP. It would be interesting and beneficial to strengthen the communicated message of the article that the authors provide more information about MB approaches, possibly exhibiting examples of balance laws and related applications (see, in particular, section II, where the concept of "hybrid models" is introduced, and section V).
  2. Related to the above question, the authors are encouraged to elaborate the concept of "virtual laboratory" which seems to be the rationale behind SS (again, the right place for this elaboration appears to be section II).
  3. The term "observer" is ubiquitous in the article but is never defined (formally). It would help the non specialized reader to have a definition of such a term.
  4. Page 5, line 167: which methods are used for linearization? Newton's method? Others? More information is needed.
  5. Page 5, line 187: how is the Lipschitz constant actually determined (or estimated)?
  6. In Table 1 reference is made to the terms "Asymptotic" (not "Asymptotical") stability and "Global" stability. What is the difference between these two kinds of stability?
  7. Page 9, line 321: the use of "the sign of the difference between measured and estimated values" to design an error estimator is not clear and deserves more information.
  8. The GDM introduced in section VI appears to be a system of ordinary differential equations in which parameters and variables depend on the sole time variable. It would be interesting to understand whether the GDM approach has been (already) or may be extended to account for the spatial variation of  the considered quantities. This would lead to a partial-differential GDM approach.
  9. Page 12, line 439: this line needs to be properly connected to the remainder of the text.
  10. A final question concerns with the numerical schemes that are typically adopted to run a SS. it would be interesting if the authors provide a short description of the principal approaches (and/or existing packages used for simulation).
  11. Page 16, line 484: in the definition of phi_m it is not clear whether the "." between Y and phi is a multiplication operator.

Author Response

Dear Editor,

We would like to thank all reviewers for their very valuable comments and suggestions. Taking into account all of them, we believe that the quality of the article has been improved.

The “Track Changes” function is used during the paper revision process. Additionally, all improvements are presented in red colour in order to be easily visible to the editors and reviewers in “Simple markup” mode.

Following are the detailed answers, point by point of each of the reviewers.

Answers to Reviewer 2

  1. To this Reviewer's opinion, the most significant aspect of this manuscript is the acknowledgement that mechanistic-based (MB) approaches may be effectively adopted to design software-sensors (SS) for BTP. It would be interesting and beneficial to strengthen the communicated message of the article that the authors provide more information about MB approaches, possibly exhibiting examples of balance laws and related applications (see, in particular, section II, where the concept of "hybrid models" is introduced, and section V).

With the answers to the questions below, I believe that additional information is given about the model-based methods. Regarding the applications, there is a great variety due to the available input information and the target product of the specific biotechnological process. Such examples can be found in the references. If they are offered in the text, it will significantly increase the volume of the article and will become a monograph.

  1. Related to the above question, the authors are encouraged to elaborate the concept of "virtual laboratory" which seems to be the rationale behind SS (again, the right place for this elaboration appears to be section II).

Regarding the term "virtual laboratory", it is not the subject of this article. We are currently developing an interactive software system for training in modeling and control of bioprocesses. It can be considered as a stage in the development of a virtual laboratory and will be the subject of other publications.

  1. The term "observer" is ubiquitous in the article but is never defined (formally). It would help the non specialized reader to have a definition of such a term.

Please pay attention to the following paragraph of the article:

„In the literature [1], it is accepted that software sensors for estimating process state variables are defined as state observers, while those for estimating kinetic parameters of the model are defined as parameter estimators.“

  1. Page 5, line 167: which methods are used for linearization? Newton's method? Others? More information is needed.

The following explanation is included in the text:

“…In the study of dynamical systems, linearization is a method for estimating the local stability of an equilibrium point of a system of nonlinear differential equations or discrete dynamical systems. Usually, a linearized approximation of the nonlinear model by a Taylor series approximation around the point of interest (equilibrium state, current estimate, etc.) is applied [1]. This is the so-called "linearized tangent model". As the linearization process leads to errors in the nonlinear system due to the calculation of the Jacobian matrix and therefore a decrease in the accuracy of the estimate, the unscented Kalman filter (UKF), central difference Kalman filter (CDKF), square-root unscented Kalman filter (SRUKF) are proposed in the literature [38]  for solving this problem. These methods are based on similar ideas and belong to a class of approximate nonlinear filtering methods (sigma point Kalman filter (SPKF) method) based on the Gaussian distribution.  In the UKF method [103], several sigma points for nonlinear systems to obtain second-order accuracy are used

Observers for linear time vatying systems

In some cases, the nonlinearity of the processes is overcome by modeling them as linear time-varying systems. In these models used in SS synthesis, linearity is achieved by considering and estimation some parameters as unknown time-varying ones [52, 59]. More detail this approach will be discussed in Sections V and VI. ..”

  1. Page 5, line 187: how is the Lipschitz constant actually determined (or estimated)?

The following explanation is included in the text:

“Observers, based on Lipschitz systems ensure convergence estimates by selecting the value of the observer tuning parameter so that the Lipschitz constant meets a specific condition. The Lipschitz constant is the maximum ratio between variations in the output space and variations in the input space of a function and thus is a measure of sensitivity of the function with respect to input perturbations [21]. A number of global optimisation algorithms rely on the value of the Lipschitz constant of the objective function [102]. Various approaches to its estimating are known, including those based on apriori knowledge of the particular process, Lipschitz optimization without Lipschitz constant, etc. [4, 106, 67].  The solutions, obtained independently of the prediction task, show a trend for noise sensitivity.”

  1. In Table 1 reference is made to the terms "Asymptotic" (not "Asymptotical") stability and "Global" stability. What is the difference between these two kinds of stability?

The term Asymptotic is included instead Asymptotical.

Regarding the difference between asymptotic and global stability, I would give the following explanation (references):

Definition The equilibrium state x = 0 is asymptotically stable (a.s.) if there exists a positive constant ε > 0 such that, if ||x(0)|| < ει, then : lim||x(t)|| = 0 under t→0

Theorem (Direct Lyapunov's method) If there exists a positive constant ε > 0 such that, if ||x(0)|| < ει , there exists a Lyapunov function for the nonlinear system, then x≡ 0 is an asymptotically stable equilibrium state.

Definition The equilibrium state x≡ 0  is exponentially stable if there exist three positive constants ε1, C1, C2 such that, for all ||x(0)|| < ε1 , the solution x(t)  is bounded as follows :

IIx(t)II< C1 exp (-C2t)IIx(0) II      Vt>0

The exponential stability implies asymptotic stability

Theorem. If the equilibrium state x = 0 of (A2.1) is exponentially stable, then it is globally stable.

This means that, globally stability implies exponentially stability

The following explanation is included in the text:

“The stability can be local, global, exponential or asymptotic depending on the method and the structure of the model [77, 97, 98].”

  1. Page 9, line 321: the use of "the sign of the difference between measured and estimated values" to design an error estimator is not clear and deserves more information.

The following sentence is entered instead of the previous one:

„This leads to their unique properties, as the ability to generate a sliding motion on the error between the measured output and the observer output ensures that the observer in sliding mode produces a set of state estimates that are exactly commensurate with the actual system output.“

  1. The GDM introduced in section VI appears to be a system of ordinary differential equations in which parameters and variables depend on the sole time variable. It would be interesting to understand whether the GDM approach has been (already) or may be extended to account for the spatial variation of  the considered quantities. This would lead to a partial-differential GDM approach.

The following explanation is included in the text:

“GDM is described by system of ordinary differential equations in which parameters and variables depend on the sole time variable, since it has been developed for stirred tanks bioreactors. Its extansion in terms of the spatial change of the state variables has not been thoroughly studied in the general case yet. In [23, 14], an   extension related with Biochemical tubular reactors is presented.  An observer of state variables based on GDM using partial-differential equations is derived. It is applied to gluconic acid production process.”

  1. Page 12, line 439: this line needs to be properly connected to the remainder of the text.

line 439 Idea for minimal modeling of reaction rates

This is a subtitle of item VI and will be emphasized

  1. A final question concerns with the numerical schemes that are typically adopted to run a SS. it would be interesting if the authors provide a short description of the principal approaches (and/or existing packages used for simulation).

The following explanation is included in the discussion section:

“…For software sensors design, different software packages (numerical schemes) are used.  They are different depending on the used method and software environment. However, there are some modules that are common to all cases:

  • A process database creation module;
  • Module containing programs that solve the differential equations of SS and / or model used;
  • Module containing programs for tuning of SS parameters and/or model identification…”

  1. Page 16, line 484: in the definition of phi_m it is not clear whether the "." between Y and phi is a multiplication operator.

Yes, this is a multiplication operator. It will be corrected to make it clearer.

Round 2

Reviewer 2 Report

The authors have successfully answered all the concerns raised by this Reviewer. According to this Reviewer's opinion, the article, in its present revised form, can be considered for publication.